# On-microscope staging of live cells reveals changes in the dynamics of transcriptional bursting during differentiation

D. M. Jeziorska[1,4,7], E. A. J. Tunnacliffe ©[1,7], J. M. Brown ©[1], H. Ayyub[1], J. Sloane-Stanley[1], J. A. Sharpe[1], B. C. Lagerholm ©[2,5], C. Babbs ©[1], A. J. H. Smith[1,6], V. J. Buckle ©[1] & D. R. Higgs ©[1,3] ✉

Determining the mechanisms by which genes are switched on and off during development is a key aim of current biomedical research. Gene transcription has been widely observed to occur in a discontinuous fashion, with short bursts of activity interspersed with periods of inactivity. It is currently not known if or how this dynamic behaviour changes as mammalian cells differentiate. To investigate this, using an on-microscope analysis, we monitored mouse α-globin transcription in live cells throughout erythropoiesis. We find that changes in the overall levels of α-globin transcription are most closely associated with changes in the fraction of time a gene spends in the active transcriptional state. We identify differences in the patterns of transcriptional bursting throughout differentiation, with maximal transcriptional activity occurring in the mid-phase of differentiation. Early in differentiation, we observe increased fluctuation in transcriptional activity whereas at the peak of gene expression, in early erythroblasts, transcription is relatively stable. Later during differentiation as α-globin expression declines, we again observe more variability in transcription within individual cells. We propose that the observed changes in transcriptional behaviour may reflect changes in the stability of active transcriptional compartments as gene expression is regulated during differentiation.

Precise spatio-temporal transcriptional control of gene expression is required to accurately produce developmental changes within a tissue or organism. Mis-regulation of this process is frequently associated with inherited and acquired genetic disease and highlights the importance of understanding in detail how transcription is regulated. While bulk and single-cell sequencing have advanced our understanding of transcription during both normal and abnormal development, these methods only provide a static view of transcription and mRNA abundance.

Observing real-time fluctuations in nascent transcription, in individual cells throughout lineage specification and differentiation, is required to fully understand the mechanistic details of gene expression[1]. Establishing when and how transient and dynamic gene activation occurs at different times during lineage specification, differentiation, and development is, therefore, of considerable current interest.

Methods allowing the visualisation of nascent transcription in individual live cells[2,3] have shown that activation of almost all genes

[1]MRC Weatherall Institute for Molecular Medicine, University of Oxford, John Radcliffe Hospital, Oxford OX3 9DS, UK. [2]Wolfson Imaging Centre, MRC Weatherall Institute for Molecular Medicine, University of Oxford, John Radcliffe Hospital, Oxford OX3 9DS, UK. [3]Chinese Academy of Medical Sciences Oxford Institute, Nuffield Department of Medicine, University of Oxford, Old Road Campus, Oxford OX3 7BN, UK. [4]Present address: Nucleome Therapeutics Ltd., BioEscalator, The Innovation Building, Old Road Campus, Oxford OX3 7FZ, UK. [5]Present address: The Kennedy Institute Of Rheumatology, University of Oxford, Old Road Campus, Oxford OX3 7FY, UK. [6]Present address: MRC Centre for Regenerative Medicine, University of Edinburgh, Edinburgh EH16 4UU, UK. [7]These authors contributed equally: D. M. Jeziorska, E. A. J. Tunnacliffe. ✉e-mail: doug.higgs@imm.ox.ac.uk

occurs in a pulsatile manner; a phenomenon known as transcriptional bursting. The use of the orthogonal MS2 and PP7 RNA tagging systems has demonstrated that numerous gene regulatory inputs are involved in producing a variety of bursting patterns (reviewed in ref. 4). Developmental changes in transcriptional bursting in live cells have previously been studied, mostly in *Drosophila*[5–7], *Dictyostelium*[2,8,9], and *Caenorhabditis*[10], while equivalent studies have not been reported in mammalian systems. To date, such studies have followed dynamic transcriptional changes occurring over relatively short time periods (1–2 h) or via sequential snapshots at different time points. This is largely due to the difficulty of conducting live imaging experiments over long time periods (up to days) without compromising cell viability or causing photobleaching. Consequently, studying transcriptional dynamics across the course of differentiation, with the sufficient temporal resolution, is problematic, making the study of mammalian development particularly challenging.

Haematopoietic stem cells undergo lineage specification and, following commitment and differentiation, mature via a trajectory of well-defined morphological stages to produce ~1–2 million red blood cells per second. This system is accurately recapitulated by various cell-based systems[11–14] and has established a general model for addressing how mammalian gene expression is regulated during changes in differentiation and development. Early in erythroid differentiation (hereafter referred to as erythropoiesis), a ~65 kb topologically associating sub-domain (sub-TAD) containing the entire mouse α-globin locus forms[15,16]. A cluster of 5 erythroid-specific enhancers is thereby brought into close physical proximity to the α-globin promoters to regulate their expression[17]. During the subsequent differentiation and maturation, including ~4 cell divisions, globin gene transcription increases and eventually, each erythroid cell accumulates up to ~10,000 molecules of α-globin RNA[18]. However, the mechanisms by which different regulatory inputs associated with changes in transcriptional bursting control such gene expression in single cells are unknown.

By combining PP7 tagging of RNA transcripts and developing "on-microscope" cell staging, we were able to observe transcription dynamics of the mouse α-globin gene in real-time throughout sequential stages of erythropoiesis. We show that nascent α-globin transcription reaches a maximum between the early and intermediate stages of erythroid differentiation, preceding peak mRNA abundance and haemoglobin synthesis, and before transcription significantly declines at later stages of differentiation. The parameters used to define transcription in live cells are summarized in Fig. 1. We find that changes in RNA production are primarily determined by the fraction of time a gene spends in an active transcriptional state (ON fraction), predominantly reflected in the burst frequency rather than the amplitude of the burst. Despite general trends in increased nascent transcription and ON fraction, we observed considerable variation in the patterns of transcriptional activity, as measured by the Fano factor (a measurement of variance/mean; Fig. 1), within and between cells at all stages of erythroid differentiation. Erythroid cells showed maximal transcriptional variability (also referred to as noise) both immediately before and after the peak in nascent transcription. Increased variability in these early and late cell stages is largely explained by the occurrence of sporadic, high-amplitude bursts.

These findings suggest that the patterns of transcriptional bursting change during differentiation, and that variability in transcription is significantly reduced at the peak period of gene expression, perhaps via the establishment of a more stable interaction between enhancers and promoters within a transcriptional hub.

## Results

### Visualising α-globin transcription in live erythroid cells derived from mouse embryonic stem cells

To understand the kinetics of gene expression in individual live cells as they differentiate, we studied the α-globin genes (Fig. 2a), which are

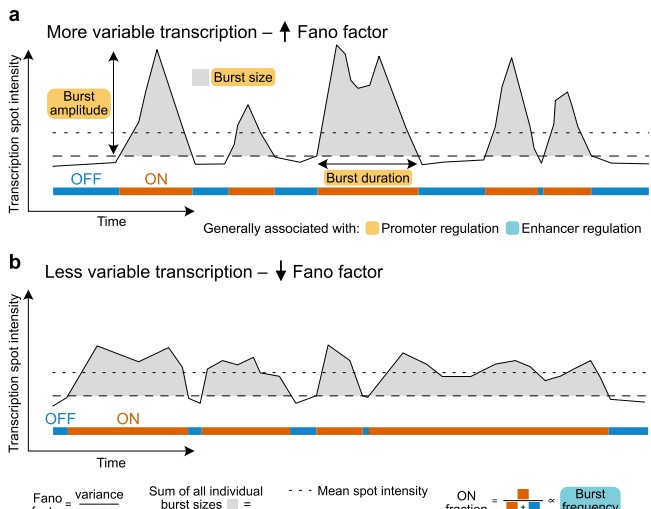

**Fig. 1 | Schematic illustrating measurable features of transcription dynamics.** Burst size changes are typically associated with promoter regulation, while burst frequency (ON fraction) is typically associated with enhancer regulation. More (**a**) and less (**b**) variable transcriptional activity traces of individual cells, with higher and lower Fano factor values, respectively, are illustrated. For highly expressed genes, burst frequency can be difficult to measure directly from spot intensity traces. Measurement of ON fraction (similar to burst fraction in fixed cells[35]) in individual cells can be used to infer changes in burst frequency.

switched on and off at specific stages of erythropoiesis. This locus provides an extremely well-characterised model that has established and illustrated many of the principles underlying mammalian gene expression[19]. To study α-globin transcription in real-time, we used the PP7 bacteriophage RNA labelling system[20]. We integrated a DNA sequence encoding an array of 24 PP7 loops into the first exon of a single allele of the *Hba-a1* gene in mouse embryonic stem (mES) cells using a recombinase-mediated cassette exchange (RMCE) strategy (Supplementary Fig. 1A) to generate *Hba-a1*-PP7 cells. Correct integration was confirmed via Southern blot and DNA FISH experiments (Supplementary Fig. 1B). To monitor changes in α-globin transcription throughout erythroid development, we used an in vitro mES cell differentiation system from which primitive erythroid cells are efficiently formed within embryoid bodies (EBs) during a differentiation period of up to 7 days[14,21,22]. In this system, at a population level, both mature and nascent α-globin transcript levels increase significantly from day 4 to day 7, alongside well-characterised changes in the expression of pluripotency and erythroid marker genes[14] (Fig. 2b, Supplementary Fig. 2). Using this system, *Hba-a1*-PP7 mES cells differentiate normally along the erythroid lineage as assessed by immunophenotyping and morphology (Supplementary Fig. 1C, D) with no evidence of cellular stress. Furthermore, these cells exhibit a normal ratio of α/β-globin RNA expression (Fig. 2c) and chromatin accessibility of the locus (Fig. 2d). Single-molecule RNA FISH (smFISH) experiments showed that transcription from both modified and unmodified alleles was largely unaffected by the introduction of PP7 repeats compared to wild-type (WT) (Supplementary Fig. 3). Therefore, labelling the endogenous *Hba-a1* α-globin gene using PP7 loops does not affect the activity of the locus or progression of erythropoiesis in differentiating mES cells.

To detect *Hba-a1* transcription in the *Hba-a1*-PP7 clone, a transgene encoding a constitutively expressed PP7 coat protein fused to GFP (PCP-GFP) was randomly integrated into the genomes of both WT and *Hba-a1*-PP7 mES cells. Clones exhibiting medium levels of GFP expression and relatively uniform expression across the population were chosen for analysis. These *Hba-a1*-PP7 + PP7 coat protein (PCP)-GFP cells differentiate normally to EBs (Fig. 2d, Supplementary Fig. 1C, E). We confirmed that a nuclear-localised transcription spot was only

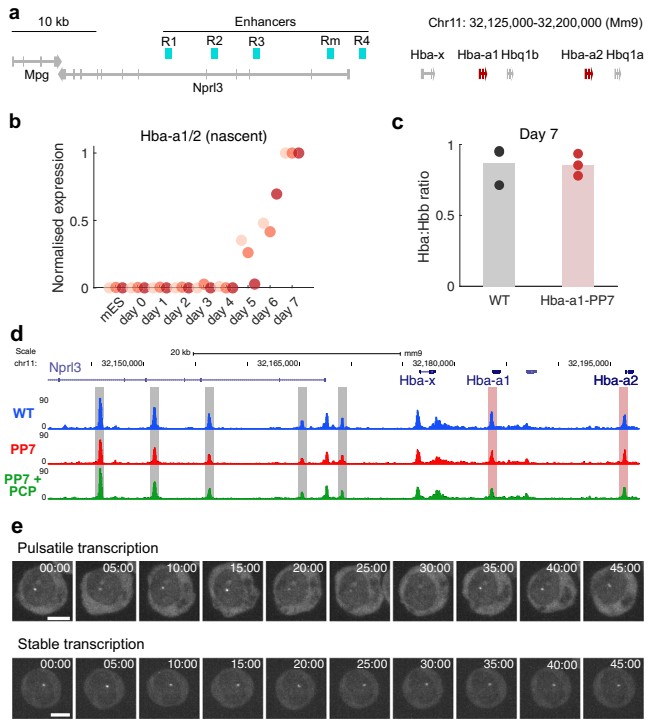

**Fig. 2 | Visualising α-globin transcription dynamics in erythroid cells using the PP7 imaging system. a** Schematic of mouse α-globin locus on chromosome 11. α-globin genes are highlighted in red, enhancers in cyan. **b** RT-qPCR time course of nascent α-globin mRNA during embryoid body (EB) differentiation. Data are normalised to Rn18s and then within differentiation time course. *n* = 3 biologically independent samples at each time point. Colours represent individual experiments. **c** RT-qPCR quantitation of α-globin mRNA abundance relative to β-globin at day 7 of differentiation. *n* = 3 biologically independent samples. **d** ATAC-seq in WT, *Hba-a1*-PP7 (PP7) and *Hba-a1*-PP7 + PCP-GFP (PP7 + PCP) cells at the α-globin locus. Grey shaded regions show known enhancers, red shaded regions show α-globin gene promoters. **e** Representative examples of nascent *Hba-a1*-PP7 transcription over time showing pulsatile or more stable gene activity in individual erythroblasts derived from day 6 EBs. Frames are a maximum projection of five z-slices around transcription spot from within the full image stack. Time in minutes. Scale = 5 μm.

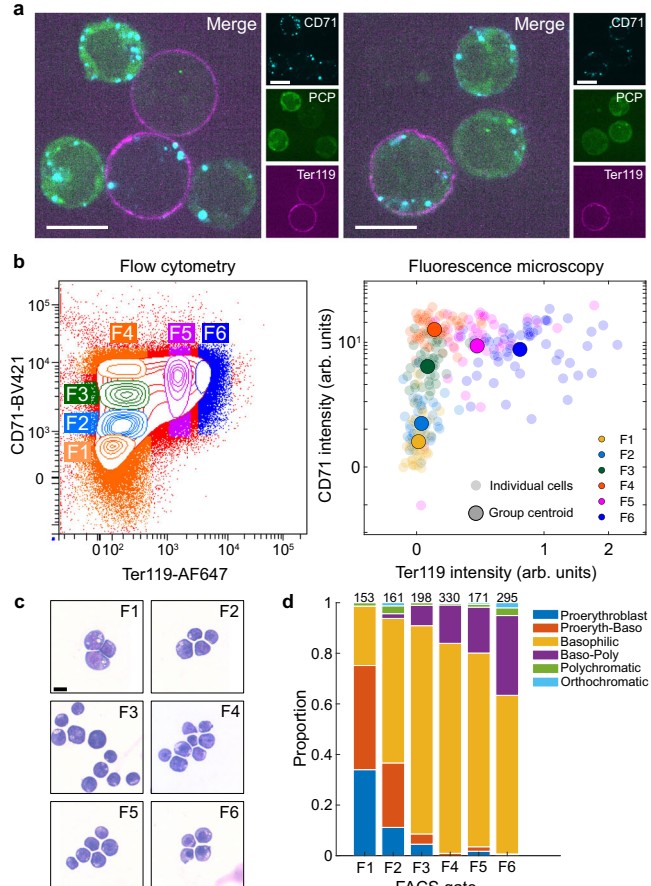

**Fig. 3 | Live-cell antibody staining enables on-microscope staging of erythroid cells. a** Example images of cells with variable levels of CD71 and Ter119 staining as well as PCP-EGFP. Single z-slices shown. Scale = 10 μm. Similar staining patterns were achieved from four independent experiments. **b** Day 7 EB-derived cells FACS-sorted (F1, F2, etc) according to CD71/Ter119 staining (left) and subsequently imaged on the microscope (right). **c** Representative images of cells from the sorted populations in B stained with May-Grünwald-Giemsa solution. Scale = 10 μm. **d** Proportion of erythroblast stages in each FACS-sorted population scored by morphology from MGG-stained cells in **c**. Numbers indicate number of cells scored. The results of this experiment were consistent with all previous staging studies using this protocol.

detected in differentiated cells containing PP7 loops integrated into *Hba-a1*, and not in WT control cells (Supplementary Fig. 4A). These transcriptional foci remain localised within the nucleus throughout the time course of the experiment (Supplementary Fig. 4B, C). However, as also observed by others in yeast[23,24], some cells exhibited cytoplasmic fluorescent foci. Such foci were excluded from further analysis (see Methods). Initial time-lapse experiments showed that the pattern of *Hba-a1* activity within individual cells derived from day 6 EBs is variable. In the course of one hour (h), some cells displayed pulsatile gene activity, and others showed a more stable, albeit still variable, pattern of transcription (Fig. 2e). This showed that α-globin may exhibit a range of transcriptional bursting behaviours in erythropoiesis.

**On-microscope staging of erythropoiesis**

While numerous studies have investigated the dynamics of gene transcription in mammalian cell culture[25–31], live-cell studies of transcription dynamics during mammalian differentiation have not been reported. To study the kinetics of α-globin transcription at different stages of erythropoiesis, we initially imaged transcription within individual cells derived from day 5, 6, and 7 EBs for 1 h with 5 minute (min) frame intervals. We observed considerable variability in the dynamics of α-globin transcription between live cells when simply stratifying by days in culture (Supplementary Fig. 5). We hypothesised that this variability was most likely due to the presence of a mixture of cells at

different stages of differentiation at each time point as a result of unsynchronized differentiation within EBs.

This problem is commonly encountered in live imaging studies of dynamic cell processes, and so, to overcome this, we stratified erythroblasts obtained from these cultures using directly conjugated antibodies[32] that recognize CD71 and Ter119, known surface markers of erythropoiesis whose levels change in a predictable manner during erythroid differentiation[11,12,33,34]. Although erythroid cells can be separated in various stages of differentiation using fluorescently activated cell sorting (FACS), we observed reduced viability of sorted cells in prolonged imaging experiments, potentially caused by an extended period of sorting-associated stress. Therefore, we developed live-cell antibody staining to stage cells throughout erythropoiesis directly under the microscope (see Supplementary Note for further discussion).

We assessed the ability of fluorescence microscopy to enable accurate cell staging during erythropoiesis. Measuring the levels of CD71 and Ter119 in a single 3D image stack (Fig. 3a) from FACS-sorted populations (F1–F6) enabled erythroid-cell staging similar to that obtained using FACS (Fig. 3b; Supplementary Fig. 6A–C), giving access to a full spectrum of erythroid differentiation states. May-Grünwald-

Giemsa (MGG) staining of the same FACS-sorted populations (F1–F6) showed CD71$^{low}$/Ter119$^-$ cells (F1) were largely comprised of proery-throblasts, while CD71$^{high}$/Ter119$^{high}$ cells (F6) represented later ery-throblast (poly/orthochromatic erythroblast) stages, with a smooth progression in the changing proportion of cells within intermediate populations (Fig. 3c, d and Supplementary Fig. 6D–F). Staining live cells by directly conjugated antibodies targeting CD71 and Ter119, therefore, enabled rapid staging of EB-derived erythroid cells into sequential fractions of the differentiation continuum by fluorescence microscopy.

### Stratifying cells into progressive stages of differentiation

Having shown that live-cell antibody staining enabled on-microscope identification of erythroblast stages, we next wanted to simultaneously monitor nascent transcription and the stage of differentiation of individual erythroid cells. We imaged *Hba-a1*-PP7 transcription for 1 h with 2.5 min frame intervals and subsequently collected single stacks of CD71 and Ter119 markers at day 6 of EB differentiation (Supple-mentary Movies 1–4). The time frame interval of 2.5 min was optimised to capture the majority of transcriptional bursts while minimising photobleaching (Supplementary Fig. 7).

To enable the stratification of these cells into progressive differ-entiation stages, we plotted cells onto a CD71/Ter119 axis (Supple-mentary Fig. 8A). We then mapped each cell onto a one-dimensional differentiation axis from the two-dimensional CD71 and Ter119 staining pattern. To stage the cells, we used an empirically defined series of curves that follow the changes in the intensity of CD71 and Ter119 markers through erythroid differentiation[14] (Supplementary Fig. 8A). Multiple curves were employed to accommodate the broad distribu-tion of CD71 staining in Ter119$^{low}$ cells. The differentiation stage of each cell was estimated from its location with respect to the nearest curve (Supplementary Fig. 8B, C). Cells were then grouped into 6 stages of erythroid differentiation (DS1–DS6) according to their positions along this continuous differentiation axis (Supplementary Fig. 8D, E). This showed a peak in the number of cells at intermediate stages of ery-throid development (Supplementary Fig. 8D) consistent with previous studies of EBs at day 6 of differentiation[14].

To validate this cell stratification approach, we compared cells at DS1–DS6 with conventionally defined erythroid precursors. We map-ped cells from the FACS-sorted populations (F1–F6) imaged by microscopy onto a differentiation axis in the same way as for DS1–DS6 (Supplementary Fig. 8F) and aligned the two axes (see Methods) to allow direct comparison between the two datasets. Subsequent superimposition of the relative proportions of erythroblast stages from F1–F6 populations (Fig. 3c, d) onto histograms of their position in differentiation (Supplementary Fig. 8Gi, ii) enabled us to approximate the differentiation stage of cells in DS1–DS6 (Supplementary Fig. 8Giii). Using these approaches, we established that DS1 represents proery-throblasts, cells at stage 3–5 (DS3–DS5) broadly correspond to baso-philic erythroblasts, and those at stage 2 (DS2) represent the transition between these two. Stage 6 (DS6) represents a mixture of later ery-throblast stages, including poly/orthochromatic erythroblasts and some basophilic erythroblasts (Supplementary Fig. 8Giii). Although not providing a perfect separation, our on-microscope staining approach enabled live imaging of transcriptional dynamics of cells at sequential stages of terminal erythropoiesis.

### Variable patterns of transcription within individual cells are seen throughout erythropoiesis

Stratifying individual cells into differentiation stages (DS1–DS6) using on-microscope analysis (Supplementary Fig. 8D, E) enabled us to test in further detail how α-globin transcription changes at specific points in erythropoiesis. For each cell analysed, we determined the average spot intensity and the fraction of time spent transcribing (ON fraction) after setting an ON/OFF threshold. For individual bursts, we recorded

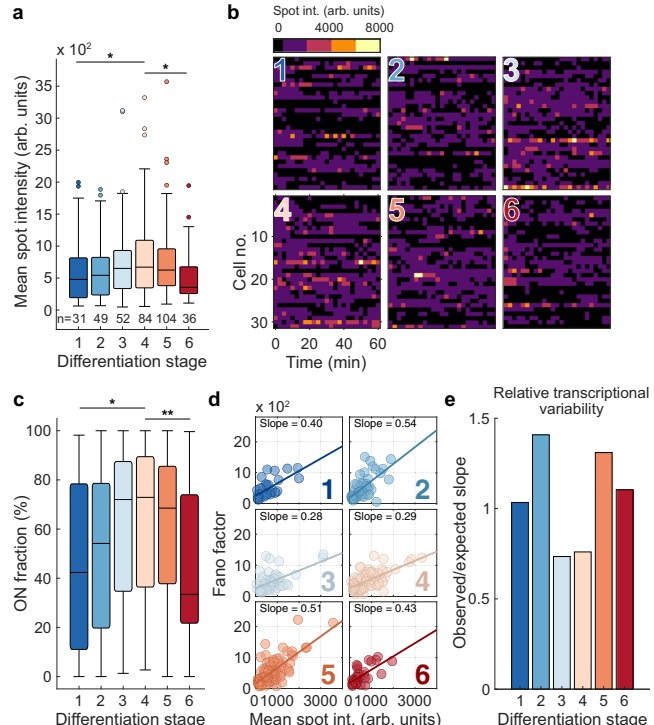

**Fig. 4 | Modulation of α-globin transcription dynamics throughout ery-thropoiesis. a** Distribution of mean spot intensities of individual cells through different stages of erythropoiesis. n values as shown from 5 biologically indepen-dent experiments. **b** Example spot intensity traces of cells within differentiation stages. Rows represent individual cells. Intensities are binned above the ON threshold of 350 arbitrary intensity units (black). **c** Distribution of the proportion of time spent in the active ON transcriptional state (above ON threshold) for each cell by differentiation stage. **d** Relationship between mean spot intensity and intra-cellular transcriptional variability (Fano factor) for individual cells across differ-entiation stages. **e** Relative levels of transcriptional variability throughout ery-thropoiesis defined by the ratio of the observed slopes of linear regression in D compared to that of all cells pooled (expected). Boxes within boxplots show median and interquartile range, whiskers show 9th and 91st percentile of dis-tribution. Statistical comparisons are two-sided Mann–Whitney U tests where \*$p < 0.05$, \*\*$p < 0.01$. *P*-values: **a** 1 vs 4: $p = 0.0496$, 4 vs 6: 0.0319. **c** 1 vs 4: $p = 0.0123$, 4 vs 6: 0.0035.

the duration and size (Fig. 1). Importantly, data were analysed using multiple ON/OFF thresholds to ensure robust results. In general, we found that cells at the mid-stages of erythropoiesis (DS3 and DS4) exhibit the highest transcription spot intensity (DS1 vs. DS4 median of mean spot intensities: $478.9 \pm 506.9$ vs $670.7 \pm 626.7$, respectively) (Fig. 4a, Supplementary Fig. 9A, B) and more frequently reach higher burst amplitude (a visual estimation of brighter colours in binned data in Fig. 4b, and raw data Supplementary Fig. 9C) in individual cells. On average, cells at stages DS3 and DS4 are active 72–73% of the time (high ON fraction), compared to only 43% and 33% for stages DS1 and DS6, respectively (Fig. 4c, Supplementary Fig. 9D). These results are con-sistent with experiments in fixed cells where, at any one time, 70–80% of cells were found to be transcribing globin genes at the peak of activation[35,36]. Together, these findings match the changes in mean transcription levels (Fig. 4a) and ~90% of the variability in mean spot intensity was explained by the fraction of time spent in the ON state (Fig. 1, Supplementary Fig. 9Ei). It has previously been suggested, based on smFISH analysis, that the fraction of cells transcribing at any one time (called burst fraction[35]) is related to burst frequency. Our use of the ON fraction is essentially a measurement of the burst fraction in live cells. While the fraction of time a gene spends actively could also be affected by the duration of bursts, measurement of individual

bursts showed that this parameter is largely invariant across differentiation (see below). Therefore, our data suggest that regulation of burst frequency is the primary determinant of the average levels of α-globin nascent transcription during erythropoiesis.

Having shown that average levels of nascent α-globin transcription (mean spot intensity) change throughout differentiation, we next wanted to determine whether the dynamic patterns of transcription within individual cells also change throughout differentiation. Fluctuations in transcriptional activity in individual cells, or "transcriptional variability", can be measured by the Fano factor (variance divided by the mean, $\sigma/\mu$), which is a measure of noise-to-signal ratio (see example traces Supplementary Fig. 9F). Using this metric we were able to assess intra-cellular variability in transcriptional activity of the gene during the imaging period (Fig. 1). In general, we found that at each stage of erythropoiesis, as gene expression increases, so does the Fano factor (Supplementary Fig. 9Fi). However, we found marked differences in the relationship between the Fano factor and the level of transcription at different times in erythropoiesis. At the peak of transcriptional activity in the population, the gradient of the linear regression between transcription (mean spot intensity) and the Fano factor within individual cells was around half that seen at the differentiation stages flanking the peak (stage 2 = 0.54 vs. stage 3 = 0.28; Fig. 4d, e, Supplementary Fig. 9G). Therefore, in the early stages of erythropoiesis (DS1 and DS2), when the α-globin genes first become transcriptionally active, the noise-to-signal ratio (Fano factor) is high relative to that which would be predicted from the level of expression (Supplementary Fig. 9F). Transcription then becomes less variable when the genes are fully active (DS3 and DS4), and again becomes increasingly variable as the genes are being switched off (DS5 and DS6) (Fig. 4d, e). These changes in the patterns of transcription, consistent with previous RNA FISH studies, suggest differences in the molecular mechanisms of α-globin activation in single cells at each stage as they progress through erythropoiesis.

## Characterising transcriptional bursting in individual cells

To investigate potential mechanistic differences in transcription at each of these stages, we analysed the relationship between different parameters of α-globin transcription dynamics (i.e. burst size and frequency, see Fig. 1) within individual cells during erythropoiesis. Regulation of these dynamics has generally been linked to the action of transcription factors (TFs) at promoters and enhancers (Fig. 1). Burst size, which can be broken down further into burst duration and burst amplitude, is thought to be regulated via the promoter[37,38]. By contrast, burst frequency is typically thought to be determined by enhancers (Fig. 1)[35,39].

We asked whether changes in burst frequency were responsible for the differences in variability during differentiation. We have shown that, throughout differentiation, the ON fraction (which is related to burst frequency[35]), is strongly correlated with mean gene activity measured from the transcription spot intensity (Fig. 4a–c, Supplementary Fig. 9Ei), suggesting that increasing burst frequency explains overall levels of α-globin RNA synthesis as erythropoiesis proceeds. By contrast, the ON fraction shows only a very weak correlation with transcriptional variability (Supplementary Fig. 9Eii), suggesting that burst frequency does not account for the observed differences in variability at different stages of differentiation.

To examine changes in burst size during erythropoiesis, we first measured the total transcriptional output (area under the curve between spot intensity fluctuations and above the ON threshold, Fig. 1) relative to the total ON duration in individual cells over a period of 60 min (Fig. 5a, b). This enabled us to assess differences in the average burst amplitude for each cell since the total transcriptional output is a product of the duration and amplitude. We estimated the average relationship between duration and amplitude across all cells using local regression and observed a clear inflection point in the gradient of this relationship (Fig. 5a). A higher gradient means increased

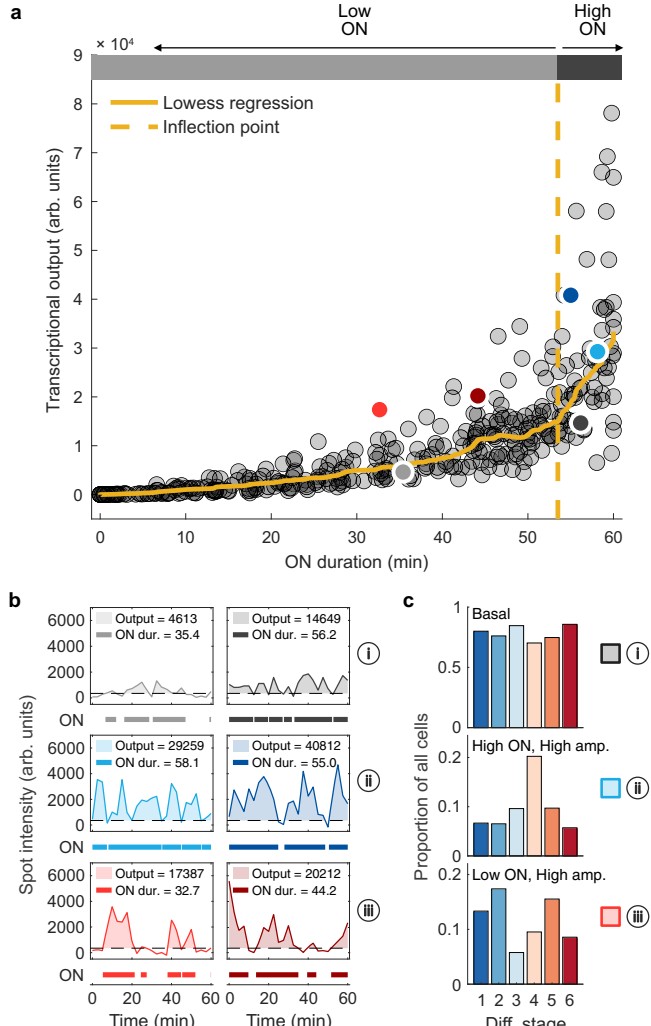

**Fig. 5 | Changes in transcriptional behaviour during erythropoiesis.**
**a** Relationship between ON duration and total transcriptional output (area below the curve and above ON threshold for entire imaging period for each cell). LOWESS regression outlines the local relationship between these variables. The inflection point marks a step-change in the gradient of the regression. Cells above the inflection point we call 'High ON' cells, while those below are 'Low ON'. **b** Example spot intensity traces for individual cells marked by coloured circles in **a**. Below each panel, thick lines indicate when intensity traces for each cell are above ON threshold (dotted lines). The measured ON duration and output (shaded area) are given for each example cell. (i) 'basal amplitude' bursting; (ii) 'high ON, high amplitude' bursting; (iii) 'low ON, high amplitude' bursting. **c** Calculated proportions of cells with different bursting behaviours in **b** at each differentiation stage (defined in Supplementary Fig. 10B, E).

transcriptional output for a given change in duration, and therefore indicates a higher average burst amplitude. Cells lying to the right of the inflection point, which is almost continuously active (which we call 'high ON' cells, active >89% of the time see Supplementary Note), therefore have an increased burst size as a consequence of increased burst amplitude. By contrast, 'low ON' cells to the left of the inflection point (Fig. 5a; active <89% of the time) have much lower average burst amplitude. In summary, these findings show that burst frequency is not primarily responsible for the observed differences in variability at different stages of differentiation. However, the burst amplitude of α-globin transcription appears to vary with the duration of time for which the gene is active (Fig. 5a). This prompted us to examine if the burst amplitude may explain the different degrees of variability that we observe at different stages of erythropoiesis.

## Coincident changes in burst amplitude and variability during erythropoiesis

To further analyse the role of burst amplitude in generating variability in transcription throughout erythropoiesis, we examined transcriptional burst traces in individual cells. While in most 'low ON' cells, expression fluctuates around a basal level of transcription (Fig. 5bi), higher amplitude bursts were observed not only in 'high ON' (Fig. 5bii) cells but also in a proportion of 'low ON' cells (Fig. 5biii). We used the distinction between 'high ON' and 'low ON' cells (Fig. 5a) to quantify the proportion of cells exhibiting these different transcriptional behaviours across the differentiation stages (Fig. 5c, Supplementary Figs. 10 and 11 and Supplementary Note). Most cells (>70%) display uniformly low burst amplitude throughout the imaging period ("basal", Fig. 5bi, ci, Supplementary Fig. 10E). The majority of cells at all stages of differentiation fall into this category (Fig. 5c, Supplementary Fig. 10E). A smaller proportion of cells (6–20%) show near-continuous activity with regular, intense bursts of transcriptional activity leading to increased transcriptional output (Fig. 5bii, cii, Supplementary Fig. 10E). We define these cells as 'high ON, high amplitude' cells. This pattern is most prominent at stages DS3-4 of differentiation when α-globin transcription reaches maximal levels (Fig. 5c). In the remaining cells, we see sporadic, strong bursts of transcription separated by periods of transcriptional quiescence ('low ON, high amplitude' cells; Fig. 5biii, ciii, Supplementary Fig. 10E). These are most prominent at stages DS1 and 2 (13–17% of cells), when transcription is starting, and also at stage DS5 (16% of cells) as the genes are being downregulated (Fig. 5c). A summary of these data is presented in Supplementary Fig. 11C. We observed that the proportion of these low ON cells exhibiting high-amplitude bursts seemed to match the trends in transcriptional variability across differentiation (Figs. 4e, 5c).

To test whether burst amplitude is indeed higher at stages DS2 and 5, when transcription is more sporadic, we characterised individual bursts in cells at all differentiation stages (Supplementary Fig. 12A). The duration of individual bursts was largely invariant across the differentiation stages (median 4.3–5.6 min) suggesting that this parameter of burst control is unlikely to be extensively regulated during erythropoiesis (Supplementary Fig. 12C). In general, high ON cells have a higher burst size for a given burst duration (indicating a higher burst amplitude) than low ON cells, in keeping with our earlier analysis (Supplementary Fig. 12D, Fig. 5a). Furthermore, the relative burst amplitude for low ON cells is highest in differentiation stages DS2 and DS5, matching the trends in noise during differentiation (Supplementary Fig. 12E, F, Figs. 4e, 5c).

Together these data suggest that increased burst amplitude in 'low ON' cells (those exhibiting sporadic transcriptional bursts) could be responsible for changes in transcriptional variability during erythropoiesis. Most importantly, the time spent in an active transcriptional state (ON fraction), most probably linked to enhancer-promoter communication, appears to be the dominant control point for α-globin transcription levels during differentiation.

## Discussion

We have studied the real-time dynamics of transcription during the process of differentiation in individual living cells. In contrast to previous studies, we have been able to determine the pattern of nascent transcription that occurs over several days as mammalian stem cells progress to fully differentiated mature cells accumulating large amounts of mRNA and protein. The key to this was our ability to accurately define erythroblast stages from a heterogeneous population under the microscope using a combination of antibodies to two well-characterised cell surface markers (CD71 and Ter119). This enabled us to define the differentiation stage of individual cells and correlate this with previously established morphological criteria. This, in turn, allowed us to develop an approach that facilitated subsequent

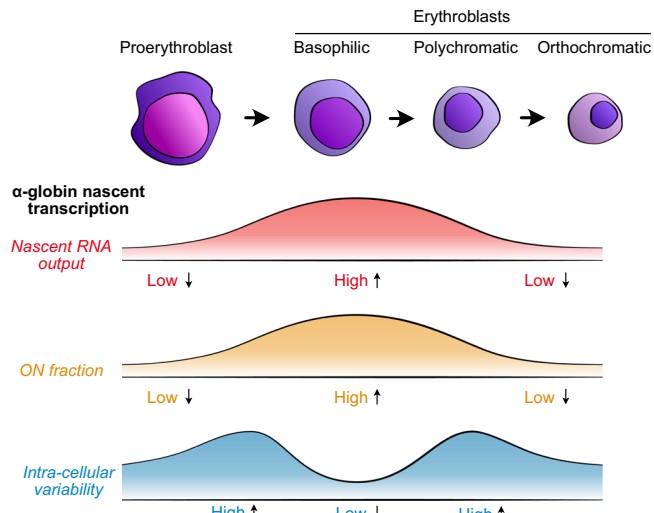

**Fig. 6 | Schematic of observed changes in transcriptional output and variability of α-globin as cells progress through erythropoiesis.** Previous work has described the sequential changes in the transcriptional, epigenetic, and chromosomal architecture of the α-globin locus throughout differentiation[19]. From this work, we find that nascent transcription is low in proerythroblast cells, peaks in the mid-stages of terminal differentiation around the basophilic erythroblast stage, before declining at later stages. This is primarily driven by the fraction of time for which α-globin is active in cells (ON fraction). We speculate that this is likely due to increased likelihood of contacts between the α-globin enhancers and promoters during differentiation as we have recently described[17]. In contrast, the relative variability of nascent transcription in individual cells is maximal either side of the peak in overall activity, and lowest when the gene is most strongly active in the population. This reduced intra-cellular variability in nascent transcription may occur as a result of formation of a more stable transcriptional hub while the gene activity is maximal in differentiation.

analysis of live imaging of transcription dynamics throughout erythropoiesis (Fig. 6).

The in vitro erythroid body culture system used here recapitulates primitive (embryonic) erythropoiesis[14], which is broadly similar to definitive erythropoiesis in terms of its core regulatory modules[40] and temporal pattern of CD71/Ter119 expression[41]. Although we have shown that changing levels of CD71 and Ter119 enable the visualisation of maturing erythroblast stages in our system, this is still a relatively coarse dissection of erythropoiesis. However, we have shown that the use of live-cell antibody staining for staging cells directly under the microscope facilitates live imaging of transcription at different stages of differentiation. As opposed to imaging of fixed cells, this approach allowed us to visualize the dynamic activity of a gene for an extended period of time, in individual cells. Given the simplicity of the method[32], this approach will greatly extend and improve the application of single-cell transcriptional dynamics in general (see Supplementary Note for further discussion).

Studying α-globin expression at well-characterised stages of erythropoiesis, we found that, consistent with fixed-cell imaging, nascent transcription (as measured by spot intensity) reaches a maximum at the intermediate stages of erythropoiesis. We also found that the main factor driving the level of α-globin transcription in any cell is the fraction of time spent in the ON state, given that the peak in both average transcriptions and ON fraction occurred in the mid-stages of differentiation, with the average likelihood of a cell transcribing at this stage being ~70% (Fig. 4, Supplementary Fig. 9). This pattern of α-globin expression is consistent with previous fixed-cell RNA FISH experiments describing transcription dynamics at several time points during blood differentiation where 70–80% of cells were found to be transcribing both α-globin and β-globin at the peak of activity[35,36].

The patterns of nascent transcription observed here in real-time were not as different at each stage of differentiation as might have been expected. Further stratification of erythroid differentiation would be expected to reveal even greater differences in patterns of nascent transcription. Each stage of differentiation included cells with three broad categories of transcriptional bursting (Fig. 5b). In most cells (>70%) at all stages of differentiation, we saw low levels of transcriptional bursting (basal Fig. 5bi). In a small proportion of cells (6–20%), we saw higher average levels of expression punctuated with intense bursts of transcriptional activity (Fig. 5bii, high ON duration, high-amplitude bursting). This is most marked at the intermediate stage of differentiation (DS4) when spot intensity and the ON fraction is at their maximum. Finally, in the remaining cells, we see sporadic bursts of transcription separated by periods of transcriptional quiescence (Fig. 5biii): these are most prominent at stages DS1 and DS2 (13–17% of cells), when transcription is starting, and also at stage DS5 (16% of cells) as the genes are becoming inactive. This suggests that between individual cells at similar stages of differentiation, there is a considerable amount of variation in the patterns of transcription. The data presented here suggest that in contrast to analyses of cell populations, which suggest transcription and the accumulation of RNA progress in a uniform manner during differentiation, the underlying process of transcription is inherently variable. Ultimately, such variability in transcription may be mitigated by post-transcriptional mechanisms, not least by changes in the relative stability of different mRNAs, leading to the accumulation of globin RNA during erythropoiesis[42].

The different patterns of transcriptional bursting observed here in real-time, as opposed to previous studies of fixed cells, suggest that the molecular mechanisms underpinning α-globin transcriptional activation may change during erythropoiesis. Based on previous studies of the transcriptional, epigenetic, and chromosomal architecture of the α-globin locus (summarised in ref. 19), we considered how these features may explain the changing dynamics of α-globin transcription. In general, enhancers have been shown to primarily mediate changes in the frequency and probability of transcriptional bursts[29,35,38]. Enhancers most frequently influence transcription by coming into close proximity to their cognate promoters[43], although exactly how this relates to transcriptional bursting is uncertain[44,45]. During erythropoiesis, the α-globin enhancers come into increasingly frequent proximity to the α-globin promoters[17] within a self-interacting domain[15]. The frequency of proximity increases in line with increased globin transcription. We have shown here that burst frequency appears to be the primary control point for α-globin transcription, suggesting that activation of the promoters by the enhancers increases the ON duration in early erythropoiesis.

It has previously been suggested that, for cells to increase transcription from a gene such as α-globin, burst frequency is first increased until a threshold is reached, above which burst size is preferentially increased[46]. Our data agree with this idea (Fig. 5), with 'low ON' cells more likely to transcribe with basal amplitude while 'high ON' cells are more likely to exhibit high-amplitude bursting. This suggests that, in the case of α-globin, the threshold proposed by Dar et al. occurs when the gene is almost continuously active, beyond which burst frequency cannot be increased further. This switch from a lower to a higher burst amplitude in high ON cells suggests α-globin can be transcribed at multiple different initiation rates as measured by the PP7 system. This phenomenon has been demonstrated for several other genes[4].

The decrease in α-globin ON duration at stages 5 and 6, could be due to mechanisms associated with chromatin condensation as cells mature into late erythroblast stages[47]. The mechanism behind the increase in the likelihood of infrequent high-amplitude bursts and transcriptional variability when the genes are not continuously active, either side of the maximum α-globin activity in the middle of terminal erythropoiesis is less clear. One component of this might be mediated via chromatin accessibility to TFs increasing as erythropoiesis progresses and then decreasing as the erythroid nucleus condenses. Recent imaging studies have highlighted TF dwell time as a key determinant of burst size in both yeast and mammalian cells with more stable binding leading to increased output[37,38]. Furthermore, in keeping with our study, a single inducible gene was shown to exhibit a range of noise-mean (Fano factor) relationships in transcription, depending on the concentration of a particular TF[48]. Recent work also suggests that the formation of transcriptional compartments, containing a high concentration of TFs may increase the efficiency of transcription[49,50]. Our observations are, therefore, consistent with the transient formation of a TF-concentrating transcriptional compartment initially giving rise to variable transcription followed by the formation of a more stable compartment producing higher levels of transcription with less transcriptional noise. Transcription again becomes variable as gene expression decreases towards the end of differentiation.

In summary, this study introduces technical advances for imaging of transcription dynamics during differentiation and development and provides fundamental insights into how dynamic patterns of transcription change during differentiation.

## Methods

### Constructs

The targeting construct to establish the acceptor site for RMCE at the α-globin locus was generated in sequential steps using λ-red-mediated recombineering. First, a Tn10-rpsL-gentR cassette flanked by AscI sites was inserted in the place of the *Hba-a1* gene sequence (coordinates 32,182,681–32,185,338 in the mouse reference genome, build mm9) in a mouse RP22-289A22 BAC clone (BACPAC Resources Centre). Subsequently, the integrated cassette and flanking 4 and 6.9 kilobase (kb) blocks of adjacent BAC-derived *Hba-a1* sequence was retrieved by gap repair recombineering into a p15A minimal vector. Finally, the Tn10-rpsL-gentR cassette was replaced in the p15 plasmid with a pgk-Hyg-TK cassette that contains a PGK promoter and linked sequence encoding a fused hygromycin phosphotransferase and HSV thymidine kinase protein flanked by loxP and lox511 sites, amplified from a modified ZRMCE vector (a kind gift from Ann Dean). Individual recombineering and cloning steps were assessed by the antibiotic selection, PCR amplification, restriction enzyme digestion, and sequencing. Thus, the final p15A-A22BAC-loxP-pgk-HygTk-lox511 construct was generated with 4 and 6.9 kb homology arms for gene targeting into the *Hba-a1* locus in order to make the RMCE acceptor site.

The *Hba-a1*-PP7 construct to be used as the donor vector for RMCE was generated in sequential steps using λ-red-mediated recombineering. First, the region of *Hba-a1* gene sequence was retrieved from BAC clone RP22-289A22 equivalent to that sequence replaced above (between coordinates 32,182,681 and 32,185,338) into a p15A minimal vector at a position with flanking loxP and lox511 sequences in the same relative orientations as in the RMCE acceptor site. The 24 PP7 repeats were synthesized (GenScript) along with *Hba-a1* coding sequences between two KflI sites at the 5′ end of the gene and subsequently exchanged by standard restriction enzyme digestion and re-ligation with the sequence in the above p15 plasmid to obtain the final p15A-loxP-*Hba-a1*-PP7-lox511 construct to be used as the RMCE donor vector.

### Cell culture

E14-TG2a.IV mouse embryonic stem (mES) cells were cultured as described previously[51–53]. Briefly, mES cells were maintained in Glasgow's MEM (Thermo Fisher Scientific, 21710025) supplemented with foetal bovine serum, sodium pyruvate, non-essential amino acids, L-glutamine, Penicillin-Streptomycin, beta-mercaptoethanol (all Thermo Fisher Scientific) and leukaemia inhibitory factor (LIF, Cell Guidance Systems, GFM200) on 0.1% gelatin-coated (Sigma, G1393) tissue-

cultured treated plasticware. A detailed protocol on mES cell differentiation to produce erythroid cells is available within ref. 14. Embryoid bodies were harvested and cells were released by trypsin-mediated disaggregation. Cell pellets were visually assessed for haemoglobinisation. Cytospins were processed by May-Grünwald-Giemsa (MGG) staining.

## Generation of Hba-a1-PP7 cell line

E14-TG2a.IV mouse embryonic stem (ES) cells were electroporated with the p15A-A22BAC-loxP-pgk-HygTk-lox511 construct to first create the RMCE acceptor site in the place of the *Hba-a1* gene. Electroporated ES cells were selected using hygromycin, and resistant clones were analysed with Southern blotting and sequencing of PCR-amplified recombined junctions to identify those arising from correct construct integration by homologous recombination. Next, correctly integrated ES cell clones were co-electroporated with the p15A-loxP-*Hba-a1*-PP7-lox511 plasmid (the RMCE donor vector) and a plasmid expressing Cre recombinase (pCAGGS-Cre-IRESpuro). Cells were selected using ganciclovir to recover those that had undergone RMCE. A ganciclovir-resistant clone was validated using Southern blot analysis with probes located at the 5′ and 3′ ends of the locus and with probes corresponding to PP7 repeats and the HygTk cassette. Targeting of a single allele was confirmed by DNA FISH. The presence of PP7 repeats in the locus was also confirmed using MinION Technology sequencing (Oxford Nanopore Technologies). The *Hba-a1*-PP7 ES cell clone was differentiated to EBs, and normal expression and chromatin landscape of the locus was confirmed by RT-qPCR and ATAC-seq.

A pPGK-PCP-GFP-IRES-Hyg plasmid (a kind gift from Jonathan Chubb) designed to constitutively express a PP7 coat protein (PCP)-GFP transgene, was then transfected into unmodified ES cells and into the *Hba-a1*-PP7 clone above. Cells were selected in hygromycin to create stable transgene-expressing cell lines. Those cell lines with a medium and relatively uniform transgene expression level as assessed by GFP fluorescence across the cell population were picked and transgene integration was validated by Southern blot analysis.

## Reverse transcription-quantitative real-time PCR

RNA was isolated from cells using TRI Reagent (T9424, Merck) before DNase treatment using DNA-free DNA removal kit (AM1906, Thermo Fisher Scientific). Superscript III First-Strand Synthesis SuperMix (11752050, Thermo Fisher Scientific) was used to generate cDNA. qPCR was performed using Fast SYBR Green Master Mix (4385616, Thermo Fisher Scientific) with primers listed in Supplementary Table 1. Data were first normalised to 18 S ribosomal RNA at each day of differentiation and then to the maximum value for that gene within the differentiation time series.

## ATAC-seq

ATAC-seq was performed as previously described[54]. Embryoid bodies were disaggregated at day 7 of differentiation, and CD71$^+$ Ter119$^+$ cells were selected for using magnetic column purification (Miltenyi). 75,000 cells were used per biological replicate. After tagmentation, the DNA was eluted using MinElute columns (28206, Qiagen). PCR indexing was performed using NEBNext High-Fidelity 2× PCR Master Mix (M0541S, NEB) and sequenced using a NextSeq platform (Illumina). After sequencing, read quality was assessed using FASTQC. Data were then aligned to mm9 build of the mouse genome using a custom-built pipeline, where PCR duplicates and ploidy regions were removed, while mitochondrial DNA was excluded during normalisation[55] (code available from https://github.com/Hughes-Genome-Group/NGseqBasic/releases).

## DNA FISH

Targeting of PP7 loops to a single allele at the α-globin locus was confirmed using RASER-FISH, a non-heat-denaturing method of DNA

FISH, as described previously[15]. Briefly, cycling cells were grown on coverslips in BrdU/C-containing medium overnight to allow incorporation during DNA replication. Cells were fixed and permeabilised before using exonuclease III (M0206L, NEB) digestion to enable resection of a single DNA strand after treatment with Hoechst 33258 and 254 nm UV light to induce nicks in the BrdU/C-containing DNA strand. Following overnight hybridisation at 37 °C and stringency washes to remove mismatched and unbound probes, hybridised probes, where applicable, were detected with appropriate antibodies and nuclei were stained with DAPI before mounting. FISH probes used were ULS550-labelled oligos (FLK 004, Kreatech Biotechnology) against the PP7 repeat sequence (AATTGCCTAGAAAGGAGCAGACGA TATGGCGTCGCTCCCT and AGCAGAGCATATGGGCTCGCTGGCTGC AGTATTCCCGGGT) while the 3′ α-globin locus probe was probe 'pA' as described in Brown et al. (2018), which was labelled with digoxygenin (DIG) and detected with sheep anti-DIG FITC (1:50 dilution, 11207741910, Roche, RRID: AB_514498) and rabbit anti-sheep FITC antibodies (1:100 dilution, FI-6000, Vector Laboratories, RRID: AB_2336218). Widefield fluorescence imaging was performed on a DeltaVision Elite system (Applied Precision) equipped with a 100x/1.40 NA UPLSAPO oil immersion objective (Olympus), a CoolSnap HQ2 CCD camera (Photometrics). Filter sets were as follows: DAPI−excitation 390/18, emission 435/40, FITC−excitation 475/28, emission 525/45, TRITC−excitation 542/27, emission 593/45. 12-bit image stacks were acquired with a z-step of 150 nm giving a voxel size of 64.5 × 64.5 × 150 nm.

## smFISH

Single-molecule RNA FISH (smFISH) was performed as described previously[56] with some alterations. Oligonucleotide probes (see Supplementary Table 2) were designed using Stellaris probe designer (Biosearch Technologies), synthesized with 3'3NHC6-modification (Eurofins Genomics), then pooled and conjugated with Alexa Fluor 594 NHS ester (Invitrogen). Cells were harvested at day 6 of EB differentiation as described above, adhered onto 22 × 22 mm glass, poly-L-Lysine coated coverslips for 20 min at 37 °C and then fixed in 4% (wt/vol) paraformaldehyde for 20 min at RT. Following fixation, coverslips were washed in PBS and then permeabilized and stored at 4 °C in 70% EtOH for a maximum of 2 weeks. After rehydration in 2× SSC with 10% (vol/vol) formamide, cells were incubated with labelled probes at a final concentration of 1 ng/μL in hybridisation buffer (2× SSC, 10% Formamide (vol/vol), 10% dextran sulfate (wt/vol), 1 mg/mL tRNA, 2 mM RNase inhibitor (RVC complex, NEB), 0.2 mg/mL BSA) in a humid chamber at 30 °C overnight, before two washes in wash buffer for 30 min at 30 °C and one in wash buffer with DAPI 0.5 μg/mL for 30 min at 30 °C. Coverslips were then mounted in Prolong Gold (Molecular Probes) that was allowed to polymerise overnight at RT, in the dark, before imaging. A widefield DeltaVision Elite system (Applied Precision) with ×100/1.40 NA UPLSAPO oil immersion objective (Olympus) and CoolSnap HQ2 CCD camera (Photometrics) was used. An Insight solid-state illumination (SSI) module (Applied Precision) was used to excite samples, with DAPI (excitation 390/18, emission 435/40) and TRITC (excitation 542/27, emission 593/45) filters used. Samples were imaged with a z-step size of 200 nm, giving a voxel size of 64.5 × 64.5 × 200 nm. Images were deconvolved using Huygens deconvolution Classic Maximum Likelihood Estimation (Scientific Volume Imaging B.V.).

## Flow cytometry

Differentiation progression was typically assessed using anti-CD71 APC (113819 Biolegend, RRID: AB_2728134) and anti-Ter119 PE (553673 BD Biosciences, RRID: AB_394986) antibodies at a dilution of 1:10,000 and 1:100 in FACS buffer (1× PBS, 10% FBS), respectively. Hoechst 33258 (1:10,000 dilution, H3569, Thermo Fisher Scientific) was used as a Live/Dead marker. Regular flow cytometry experiments were done on an

Attune NxT Analyser (Thermo Fisher Scientific) while sorting experiments were done using a FACS Aria Fusion sorter (100 μm nozzle width; BD Biosciences). FlowJo (v 10.7.1, FlowJo LLC) was used to analyse flow cytometry data. Day 7 (chosen to ensure a full range of differentiation states) EB cells for sorting were placed in a recovery medium (phenol red-free IMDM base medium, 10% FBS, 1 U/ml erythropoietin) for 1 h following disaggregation before being stained (anti-CD71 Brilliant Violet 421, 1:5000 dilution−113813 Biolegend, RRID: AB_10899739; anti-Ter119 Alexa Fluor 647, 1:1000 dilution−116218 Biolegend RRID: AB_528961) in FACS buffer and sorted based on CD71 and Ter119 signal (Fig. 3b, Supplementary Fig. 6B). A minimum of 250,000 cells were collected for each of 6 sorting gates, with half taken for cytospins and the remainder used for fluorescence microscopy. Data from cells imaged on a microscope were plotted on a biexponential scale (as is typically used for flow cytometry data) in Matlab using the 'logicleTransform' function (https://uk.mathworks.com/matlabcentral/fileexchange/68289-logicletransform-m). This transformation requires custom labelling of axes; the file in Supplementary Software 1 is provided to enable this and, therefore, full recapitulation of plots within the manuscript (using the data available in the accompanying Source Data file).

### Live imaging

Prior to imaging, embryoid bodies were disaggregated as described above, and cells were placed in a recovery medium for 4 h. Cells were then passed through a cell strainer, counted, and 500,000 cells were allowed to settle on poly-L-lysine coated 35 mm high μ-Dishes (81158, Ibidi) in fresh recovery medium. Cells were then imaged using a 488 nm laser (100 mW) with 100 ms exposure at 50% power every 10 s for 10 min for short movies, or every 2.5 min or 5 min for 1 h for long movies, with stacks of 30 z-slices sampled every 500 nm in the z-plane. For measurement of CD71 and Ter119, cells were stained for 5 min before imaging with directly conjugated anti-CD71 Brilliant Violet 421 (1:5000 dilution) and anti-Ter119 Alexa Fluor 647 (1:1000 dilution) antibodies, respectively. Sodium azide was removed from antibodies before use by diluting in 2 ml PBS and re-concentrating using a protein concentrator column (88521, Thermo Fisher Scientific) as this is critical for live imaging experiments using antibodies[32]. Multiple stacks were collected in parallel after 1 h of imaging using 405 nm (50 mW), 488 nm, and 635 nm (30 mW) lasers with 100 ms exposure and at 25%, 50%, and 50% power, respectively, as well as a brightfield DIC image stack. FACS-sorted cells were imaged in the same way but without 1 h time-lapse imaging. For nuclear labelling experiments, cells were stained with SiR-DNA (SC007, Spirochrome) at 1:1000 dilution for 1 h. Cells were imaged at 37 °C on an inverted Zeiss Cell Observer Spinning Disc system with a CSU-X1M 500 Dual Cam spinning disc unit (Yokogawa), a ×1.2 magnification camera C-mount adapter, an Orca-Flash4.0 v2 sCMOS camera (Hamamatsu), and a Plan APO ×63/1.40 NA Oil M27 objective (Carl Zeiss AG) with a final voxel size of 86 × 86 × 500 nm. ZEN Blue 2 software (Zeiss) was used to capture images. Five independent live imaging experiments were performed as described above.

### Image analysis

Fluorescence microscopy images were uploaded to OMERO[57] for storage and initial cell identification. Healthy erythrocyte lineage cells were manually identified by morphology: spherical cells lacking obvious blebs, an intact nucleus (mitotic cells were excluded), and absence of intra-cellular staining with Ter119 antibody (as this was identified as a marker of dead or dying cells). Cell centroids were marked using the 'point' ROI tool in OMERO.iviewer, and locations and images were downloaded from the OMERO server using the Matlab (Mathworks) API.

Manual quantification of spot intensities from cells at day 5, 6, or 7 of embryoid body culture (Supplementary Fig. 5) was done using

Imaris (version 9.1.2, Oxford Instruments). The intensity of transcription spots was quantified using the 'Spots' tool and calculated as the mean of pixels within an ovoid shape of $1 \times 1 \times 3$ μm in size (x,y,z dimensions, respectively), centred on the transcription spot. The nuclear background was measured within a region of size $3 \times 3 \times 3$ μm within the nucleus at the same z-position but situated away from the transcription spot in the xy plane. Corrected spot intensity was calculated by subtraction of background values from measured spot intensity. If no spot was easily visible (if the gene was inactive), measurements were taken from the coordinates of the last visible spot. Kymographs of SiR-DNA-labelled *Hba-a1*-PP7 nuclei were created in Imaris by first creating a surface for individual channels using simple thresholding. Cells maintaining active transcription spots throughout the imaging period were chosen for ease of visualisation. Single 2D image slices (xy dimensions) from each time point, centred on the transcription spot in the z-dimension, were assembled into a 3D image stack with time as the third dimension in order to demonstrate continued nuclear localisation of transcription spots during imaging.

Automated identification and quantification of transcription spot intensities were done in Matlab (version R2014a), largely as described previously[58], with minor modifications. All other analyses in Matlab were done using version R2020a. In brief, since we did not use a marker such as a fluorescent histone tag to outline the nucleus and given that some MCP-GFP foci were visible in the cytoplasm of some cells, we generated a pseudo-nuclear marker by exploiting the relative depletion of GFP signal in the nucleus compared to the cytoplasm (Supplementary Fig. 4B). Cells were analysed individually by using centroid locations to crop the original images, and a simple two-step k-means clustering algorithm was used to segment first the cell boundary, and then the nuclear boundary within this. This estimate of nuclear position was then passed with GFP image stacks to a spot detection algorithm[58] for automatic detection and measurement of transcription spots. Manual inspection and correction of spot locations were done, where necessary, to ensure high accuracy. This semi-supervised automated analysis showed good agreement with manual quantification of spot intensities in initial experiments (Supplementary Fig. 5). Measurement of CD71 and Ter119 marker levels was done by similarly cropping and segmenting a cell mask from the GFP channel, before taking the mean value in each channel of all pixels within the mask, minus an estimated background (iteratively smoothed from pixels outside the cell mask until no change is observed, as described previously[58]) for each cell. Cell size was approximated by taking the area of the cell mask in the central z-slice (centroid of the mask in the z-dimension) for each cell. Given the highly spherical nature of early erythroid cells, this proved to be a sufficiently good estimate for our studies.

For smFISH experiments, cells were scored manually for the number of active α-globin genes by counting the number of bright foci within the nucleus. Only cells for which the entire nucleus was contained within the image stack were counted. Cells in which two nascent transcription foci from a duplicated chromosome were visible (two signals in close proximity) were counted as a single active locus.

MGG cytospins were scored manually according to changes in size, colouration and texture of nuclei and cytoplasm of cells using the 'Cell Counter' plugin in Fiji[59] after image names were randomised. Erythroblast counts for each image were then unblinded and collated according to FACS-sorted populations (F1–F6).

### Data analysis

Short movies (10 min; Supplementary Fig. 7) were used to establish an optimum frame rate for longer-term imaging of transcription dynamics (1 h). Transcription spot intensities were extracted from images as described above. The data were then subsampled at progressively increasing frame intervals for each cell to simulate the use of different experimental imaging sampling intervals. We

wanted to establish the optimum sampling interval by which to capture all transcriptional bursting events while also minimising light input and associated photobleaching. To do this, we estimated the amount of information loss which occurs with increasing frame rate by calculating the root-mean-squared error (RMSE) when comparing the raw data (sampled every 10 s, $y_{raw}$) to subsampled data ($y_{sub}$) for each cell:

$$\text{RMSE} = \sqrt{\frac{\sum_1^n (y_{sub} - y_{raw})^2}{n}} \qquad (1)$$

A lower RMSE indicates a reduced error (information loss) of subsampled compared to raw data. A sampling interval of 150 s was found to be a good compromise to minimise both light input and information loss.

Differentiation progression for cells imaged by microscopy was estimated using the measured intensities of CD71 and Ter119 staining (Supplementary Fig. 8). Cells undergoing embryoid body differentiation exhibit known changes in the levels of these two markers during differentiation progression, from CD71⁻/Ter119⁻, to CD71⁺/Ter119⁻, to CD71⁺/Ter119⁺[14] (Supplementary Fig. 6B). We used this knowledge to measure the progression of each cell through this differentiation space. To do this, we empirically defined a series of curves that followed this pattern of changes in CD71 and Ter119 markers. Several curves were used in order to capture the full range of potential trajectories through this CD71/Ter119 axis, given that a range of CD71 intensities were observed in Ter119ˡᵒʷ cells. These curves were then discretised into 10,000 equally spaced points using the 'interparc' function in Matlab. We measured the distance between each cell and every point on these curves, found the point closest to each cell, and calculated the proportional distance of that point along its curve as a fraction of 1. This value was taken as a proxy for the extent of differentiation progression for each cell.

Having defined a series of differentiation stages (DS1–DS6) from time-lapse imaging long movies, we wanted to estimate which erythroblast stages were most likely to represent these stages. To do this, we used microscopy measurements of CD71 and Ter119 intensity to estimate differentiation progression (as above) of day 7 EB FACS-sorted cell populations (F1–F6) (Fig. 3b, Supplementary Fig. 8F). Unsorted day 7 cells (Supplementary Fig. 6B) were also included to ensure enough cells to enable comparison to the long movie time-lapse dataset. To allow comparison between the time-lapse dataset and the FACS-sorted dataset, we used regular fluctuations in the size of cells along the differentiation axes (from 0 to 1) to align the datasets in differentiation. Following this alignment, the coordinates of cells in differentiation should be equivalent for DS1–DS6 and F1–F6. Given that we assessed the identity of erythroblast stages in F1–F6 (Fig. 3c, d), and now knowing where these cells were on the differentiation axis, we could approximate the cell identity in time-lapse imaging differentiation stages. Firstly, the distribution of FACS-sorted cells on the differentiation axes was plotted separately for each sorted population F1–F6 (Supplementary Fig. 8Gi). Secondly, the known proportions of erythroblast stages for each of these populations as assessed by morphology analysis (Fig. 3c, Supplementary Fig. 8Gii) were overlaid onto these histogram distributions from least to most differentiated (proerythroblast, basophilic erythroblast, polychromatic erythroblast, orthochromatic erythroblast; Supplementary Fig. 8Giii). Finally, using the aligned coordinates of the time-lapse imaging differentiation stages (e.g. DS1 at 0 to -0.1) on this differentiation axis, we assigned erythroblast cell identity to each of these groups by visual assessment of the erythroblast stages across F1–F6 at these coordinates (Supplementary Fig. 8Giii).

Confidence intervals for the median of mean spot intensities for cells at each differentiation stage (DS1–DS6) were calculated using bootstrapping (Supplementary Fig. 9B). For each differentiation stage, the distribution of mean spot intensity values was randomly sampled

with replacement ($n = 31$, to match the lowest number of cells in any differentiation stage, DS1) 10,000 times and median values determined. Limits of 95% confidence intervals were calculated as the 2.5th and 97.5th percentiles of this bootstrapped distribution.

The threshold above which α-globin was considered to be active or 'ON' was defined empirically by visual inspection of images to be 350 arbitrary intensity units. However, at each analysis stage, care was taken to test a number of thresholds around this value to ensure that conclusions were consistent regardless of which ON/OFF threshold was chosen. The ON fraction was calculated for each cell as the fraction of time the gene spends in the ON state as a proportion of the total imaging period. To more precisely estimate time spent above the ON/OFF threshold, we used linear interpolation of individual transcription traces. Duration of individual bursts was measured similarly, but only 'complete' bursting events were included, where both the start and end of a burst were below the ON/OFF threshold. The transcriptional output of an individual cell was calculated as the area above the ON/OFF threshold and below the fluctuating (interpolated) transcriptional activity trace summed across the imaging period. This is effectively the summed burst size for all bursting events (including those 'incomplete' bursts). Burst size for individual bursts was measured similarly but again only included 'complete' bursting events. Calculating the number of complete bursts identified across a large range of ON/OFF thresholds demonstrated a peak centred around our empirically defined intensity threshold of 350, further supporting the use of this threshold estimate (Supplementary Fig. 12B).

In order to categorise cells according to differing transcriptional behaviours, we first distinguished cells according to the length of time spent in the active state (ON duration). We did this based on the relationship between the ON duration and transcriptional output, for which we used robust LOWESS local regression ('rlowess' in Matlab 'smooth' function) to model. A clear inflection point suggested differences in burst amplitude at high ON duration (see Supplementary Note), and we, therefore, categorised cells as 'High ON' or 'Low ON' according to whether their ON duration was above or below this inflection point, respectively (Fig. 5a). We then categorised cells according to the amplitude of transcriptional bursts displayed over the imaging period as we noticed many cells consistently showed a low or 'basal' level of burst amplitude while some exhibited much higher burst amplitude (Fig. 5b). We again used the relationship between ON duration and transcriptional output to define a threshold above which cells would be classed as exhibiting 'high amplitude bursting' on average across the imaging period (Supplementary Fig. 10B). Firstly, we estimated the relationship between ON duration and transcriptional output for a basal amplitude bursting regime by extrapolating from the local regression described above for Low ON cells only, as the majority of these cells exhibited basal amplitude bursting (yellow curve, Supplementary Fig. 10Bi). Fitting a quadratic curve to the Low ON regression data points allowed us to estimate the transcriptional output for basal amplitude bursting in High ON cells. To categorise cells according to whether they exhibit 'basal' or 'high' amplitude bursting, we then wanted to measure how far each cell deviates from basal amplitude bursting, in terms of its transcriptional output. If the transcriptional output of a particular cell is very close to what would be expected from basal amplitude bursting with a certain ON duration, then this deviation would be small. In contrast, for cells exhibiting high-amplitude bursting, and therefore with a high transcriptional output for a particular ON duration, this deviation would be much higher. To more easily define a single threshold value with which to identify high-amplitude bursting cells, we first calculated the residual between the transcriptional output of each cell and the basal amplitude estimate of this value for the same ON duration (Supplementary Fig. 10Bii). We then normalised these residuals by the ON duration, to account for the increased time available for

higher ON cells to deviate from a basal amplitude bursting regime (Supplementary Fig. 10Biii). We call this metric the 'basal amplitude deviation score'. Finally, we used the distribution of the basal amplitude deviation scores to set the high-amplitude bursting threshold. In Low ON cells, as expected, we found a close-to-normal distribution of deviation scores around a value of 0 (meaning the deviation of most of these cells from a 'basal amplitude bursting' behaviour was small), with a skewed tail of high-amplitude bursting cells. Setting the threshold as one standard deviation away from the median enabled the convenient separation of basal and high-amplitude cells. Cells were then classed into three groups by using the High/Low ON and High/Basal amplitude thresholds to define three different transcriptional behaviours: 1. Basal amplitude (both Low and High ON), 2. High ON, High amplitude, 3. Low ON, High amplitude. A similar analysis was performed using the Fano factor instead of the basal amplitude deviation score (Supplementary Fig. 10D), but we found this to be less suited to the task of segregating cells by burst amplitude (see Supplementary Note).

## Data availability

The data that support this study are available from the corresponding author upon reasonable request. The ATAC-seq data generated in this study have been deposited in the Gene Expression Omnibus (GEO) database under accession code GSE189474. All image data associated with this work are stored in OMERO and will be available from the authors upon request. Source data are provided with this paper.

## Code availability

Image analysis was primarily done using published Matlab code[58] (see http://www.ucl.ac.uk/lmcb/sites/default/files/Corrigan2016MatlabFiles.zip). ATAC-seq data were aligned to the mouse mm9 genome (UCSC Genome Browser, July 2007 NCBI37/mm9) using a custom analysis pipeline[55] (available from https://github.com/Hughes-Genome-Group/NGseqBasic/releases). All other code for this work is available from the authors upon request.

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

## Acknowledgements

We are very grateful to Jonathan Chubb and all members of the Higgs lab for helpful comments on the manuscript; Damien Downes for assistance with submission of data to the Gene Expression Omnibus; Dominic Waithe for initial suggestions for image analysis; Craig Waugh and the Flow Cytometry Facility at the Weatherall Institute for Molecular Medicine (WIMM) for help with FACS sorting experiments; the Computational Biology Research Group and Wolfson Imaging Centre at the WIMM for bioinformatic support and use of microscopy facilities and associated services. This work was supported by Medical Research Council grants MC_UU_00016/4 (D.R.H.) and MR/T014067/1 (D.R.H.).

## Author contributions

D.H. secured funding for the project. D.J. and D.H. conceived the study. D.J., E.T., J.B., H.A., J.S.-S., J.S. and A.S. performed experiments and provided reagents. D.J., E.T. and J.B. analysed the data. B.L., C.B., A.S., V.B. and D.H. provided supervision and expertise. E.T. and D.H. wrote and revised the manuscript in consultation with D.J. All other authors provided comments on the manuscript before submission.

## Competing interests

D.M.J. is co-founder of Nucleome Therapeutics. The remaining authors declare no competing interests.
