## [Peer Review File · Nature Communications]

REVIEWER COMMENTS

Reviewer #1 (Remarks to the Author):

In their manuscript Jeziorska et al. aim to characterize transcriptional dynamics from the hemoglobin alpha gene, at the resolution of single live cells, during erythroid differentiation from mouse embryonic stem cells. Gene recombination tools were used to tag a single allele of the Hba-a1 gene with PP7 stem loop sequences, which enabled visualization of dynamic real-time gene expression. The authors develop a scheme to identify erythroid precursors at various stages of differentiation by live cell labeling using fluorophore-conjugated antibodies directed against two different surface antigens. Even after classifying individual cells to different states of differentiation, much variability was identified for several transcription parameters, such as on time, both within classes and between states of differentiation. Nevertheless, transcriptional features were identified that roughly aligned with stages of early, mid and late erythroid differentiation.

The goal of studying transcriptional dynamics in a continuous fashion through the course of differentiation in mammalian cells is of significance and laudable. The present study did not quite meet that objective since individual cells were analyzed for relatively brief periods, less than 2h, for an erythroid differentiation process which occurs over several days. Further, the live cell microscope protocol used for classification of cells to progressive differentiation states were not especially robust, deriving states that overlapped extensively (e.g., Fig. 3B). These drawbacks may have contributed to overlapping and highly variable transcription patterns observed between assessed differentiation states. Still, the results of this manuscript's provide new information on specific transcriptional parameters that may be altered for Hba expression during differentiation, in comparison to those previously inferred by fixed cell studies (as in Bartman et al.).

Overall, the paper is well written, the figures are beautiful, and the experiments are well performed and controlled.

Some points that should be addressed:

1. One concern is the PP7 stem loops were placed in the 5' end of the gene rather than in an intron. Previous reports have shown this can interfere with translation (see PMID: 26017315 and PMID: 2108146), which may stress cells and alter their behavior. Also, since only a single allele was tagged, it is possible the tagged allele did not behave normally and instead the other allele compensated. To test this, it would be important to do RNA FISH to confirm the tagged and untagged alleles behave similarly.

2. The authors state: "we have shown that the use of live-cell antibody staining for staging cells directly under the microscope facilitates live imaging studies over a prolonged period of differentiation" (Discussion p.2, pg13). This seems like an overstatement given their study essentially just captured temporal snapshots of the process as with previous reports. Indeed, FACS sorting coupled with smFISH may have provided similar overall results.

3. Related, the manuscript would be improved considerably if the authors included data tracking single transcription sites for longer periods of time, preferably from one stage of differentiation to another. Given the authors' interesting observation that burst amplitude ramps up at intermediate stages of differentiation, it would be nice to see how this plays out in a single cell. Note the imaging wouldn't have to be continuous. For example, a single cell could be imaged continuously for 30 minutes, then imaged very infrequently (or not at all) for a relatively long period of time, then imaged continuously again for another 30 minutes...

4. The authors develop a model based on their observations in Figure 6, but do not attempt to test the model. The study as it stands is just observational. It would be nice to perform some sort of perturbative experiments to test a model in a meaningful way. For example, siRNA could be used to knock down factors known to promoter enhancer/promoter crosstalk, such as P300 or cohesion, or activate gene expression, such as a specific transcription factor. Such perturbative experiments would yield insight into the mechanism of bursting at this gene, which would be very informative.

4. The data are not particularly convincing that live cell microscope imaging meaningfully supports identification of 6 layers of differentiation strata, especially since at least 3 categories rely solely on a narrow range of CD71 fluorescence intensity. This range of CD71 expression could encompass noise in CD71 expression itself, unrelated to differentiation status.

5. The data for Fig. 5D is difficult to interpret relative to that presented in Fig. 4D. A similar presentation to that for Fig. 4D, perhaps with additional separate panels for 'low ON' vs 'high ON', would make it easier to evaluate these data.

6. The 'Ter19 intensity' x-axis for Fig. 3B and other figures require additional numeric labeling.

Reviewer #2 (Remarks to the Author):

The manuscript submitted by Jeziorska, Tunnacliffe, et al. is a major step forward in our understanding the mechanisms by which genes are switched on and off during differentiation. Current models of gene transcription based on the observation of fixed cells has shown that even the most active genes are not continuously transcribed based on the observation of both active and inactive cells within a population. This has led to the conclusion that transcription is sporadic and interspersed with periods of inactivity. In this paper Jeziorska, Tunnacliffe, et al. used a novel onmicroscope analysis of mouse α -globin transcription in living ES cell derived erythroblasts carefully characterized by surface markers that define stages of erythropoiesis. The procedure involves PP7 tagging of RNA transcripts and developing "on-microscope" cell staging using antibodies to CD71 and Ter119 to allow a measure of the activity of an α -globin gene in a cell with a defined surface marker phenotype in what turned out to be a nonsynchronized population of cells. Jeziorska, Tunnacliffe, et al. found that levels of α -globin transcription were highest in cells in the mid stages of erythroid maturation. In these cells, the fraction of time a gene spends in the active transcriptional state is high compared to the earlier or latter stages of erythroid maturation, where α -globin transcription is lower. Individual actively transcribing cells showed considerable variation in α -globin transcription within and between cells at all stages of erythroid differentiation. Maximal transcriptional variability was most frequently observed in the early and late stage cells. The authors propose that transcriptional activity in the form of "bursts" changes during differentiation. At the peak stages of α -globin transcription, the variability in burst levels and the periods of inactivity are much lower than was seen in the early and later stages. The authors speculate that this may reflect changes in the stability of the interaction of enhancers and promoters at each stage. This paper has many strengths. The question of transcriptional regulation is fundamental to almost every field of biology, and readers in any of these fields are likely to be influenced by this work. The authors are commended for keeping the narrative focused on α -globin as a model locus, rather than shifting the focus to the erythroid system. The experiments are exquisitely controlled, and the authors go to great lengths in the text and supplemental

materials to make sure that the techniques and analysis tools are explained. The figures are concise and informative. It was a pleasure to review a paper that did not contain 8 panel figures.

Speaking directly to novelty, this is definitely new and important work. Although the results are consistent with previous models, there are critical and important differences between these studies and previous work. Firstly the experiments were done in live cells and were imaged over time as opposed to extrapolation from populations of fixed cells. The ability to "stage" the cells at specific points in differentiation are a major step forward and show how transcription is clearly regulated very precisely.

As far as weaknesses go, I could find none that would require additional experiments. Perhaps a microscopist familiar with the problem of photobleaching and signal detection could comment on the reliability of the observations. My only minor suggestions are that the authors make some mention that they are studying embryonic erythropoiesis as opposed to the more familiar definitive wave. While it is unlikely that there are significant differences, the previous work was mostly done in definitive erythroblasts. Secondly, while I enjoyed the model of enhancer/promoter interactions being altered, the results really do not look at either regulatory region. I would favor a model slide focusing on the activity of the gene itself and leave the promoter/enhancer interactions for future work.

Reviewer #3 (Remarks to the Author):

This manuscript studies the transcriptional activity of the alpha-globin gene during erythroid differentiation, using single cell live imaging of endogenously PP7-tagged nascent transcripts. Using live antibody staining, they classify differentiating cells in different stages to dissect changes in transcriptional dynamics during differentiation.

They found that the main driver of changes in transcriptional output is changes in burst frequency, will differences in burst amplitude mainly contribute to cell-to-cell variability in alpha-globin transcription. They suggest that at some stages of differentiation, high burst amplitude combined with low burst frequency contribute to higher noise.

Overall, this work is very well-done and the study of transcriptional dynamics during differentiation is novel. I do not have any major comments on the quality of the experimental results and analysis. The authors also suggest a number of interesting mechanisms that could explain how burst frequency and amplitude are modulated (promoter-enhancer contacts, formation of hubs, chromatin accessibility).

Main comments:

The major weakness of this paper is the use of a single approach (PP7 live imaging) to study transcription. While this clearly allowed the authors to perform a fine dissection of transcriptional dynamics during differentiation, the mechanisms that lead to these changes remain speculative. Thus the novel insights provided by this manuscript on our understanding of transcriptional dynamics are somehow limited. In my opinion, the authors should substantiate the mechanistic hypotheses they make by experiments probing promoter-enhancer contacts / chromatin accessibility / hub formation, and link these to the observed changes in dynamics. This could be done either using single cell approaches (DNA FISH to study enhancer-promoter proximity in combination with their antibody stainings, etc.) and/or using population-based approaches (4C, ATAC-seq) on sorted cell populations.

Minor comments:

1. Parameters for image stack acquisition should be completed with number and thickness of slices

2. Extended data Fig.2: The data is normalised, but for a number of panels the max value is not 1 (e.g. Runx1, Nanog), this should be corrected

3. Extended data Fig.3B: Why showing two examples of the same ?

We are grateful for the careful review of the manuscript and for the opportunity to respond the few points that were raised.

Reviewer #1 *“the results of this manuscript’s provide new information on specific transcriptional parameters that may be altered for Hba expression during differentiation, in comparison to those previously inferred by fixed cell studies (as in Bartman et al.). Overall, the paper is well written, the figures are beautiful, and the experiments are well performed and controlled”.*

1. *One concern is the PP7 stem loops were placed in the 5’ end of the gene rather than in an intron. Previous reports have shown this can interfere with translation (see PMID: 26017315 and PMID: 2108146), which may stress cells and alter their behavior.*

Since there are four functional alpha genes and we have tagged just one, it seems unlikely that this would cause significant stress to the cell. In humans in which one alpha gene is deleted, there are minimal changes in the haematological parameters in the peripheral blood (Harteveld and Higgs, Orphanet Journal of Rare Diseases 2010). In this study, in mouse, we showed that the tagged cells differentiate normally and accumulate haemoglobin as wild type cells and therefore we found no evidence of significant cellular stress as now specified in the text (page 5).

2. *Also, since only a single allele was tagged, it is possible the tagged allele did not behave normally and instead the other allele compensated. To test this, it would be important to do RNA FISH to confirm the tagged and untagged alleles behave similarly.*

It has previously shown in both human and mouse that when one allele of the alpha globin locus is compromised (e.g. by deletion of both globin genes from one allele) there is no compensatory increase in expression from the other allele (Harteveld and Higgs, Orphanet Journal of Rare Diseases 2010, Paszty et al Nature Genetics 1995). The alpha/beta globin RNA and protein synthesis decreases approximately in accordance with the number of alpha genes that are deleted. In this study, using qPCR analysis we found that the alpha/beta globin RNA ratio in the tagged line was ~1 and indistinguishable from the non-tagged cells.

To address this more formally, as the referee suggested, we used single molecule FISH studies and showed that using this orthogonal approach, transcription from the modified and unmodified alleles was largely unaffected. The new data addressing this are shown in Extended Data Fig. 3 and we comment on this on page 5 of the manuscript.

2. *The authors state: “we have shown that the use of live-cell antibody staining for staging cells directly under the microscope facilitates live imaging studies over a prolonged period of differentiation” (Discussion p.2, pg13). This seems like an overstatement given their study essentially just captured temporal snapshots of the process as with previous reports. Indeed, FACS sorting coupled with smFISH may have provided similar overall results.*

While the reviewer is correct that smFISH and PP7-based live imaging of transcription are similar in their ability to monitor nascent transcription in some respects, there are some important distinctions to highlight. Given that smFISH is a fixed-cell method providing a “snapshot” of transcription, we would only have been able to *infer* bursting patterns using this method, whereas in this study we have *directly shown* transcriptional bursting of alpha-globin occurring in real-time. Furthermore, by live imaging over the course of 60 minutes we were

able to identify differences in the *intra*-cellular variability (over time) of nascent transcription in individual cells which would not have been possible with a fixed-cell method like smFISH. To make our statement about this approach more accurate we have revised the text (page 13).

3. Related, the manuscript would be improved considerably if the authors included data tracking single transcription sites for longer periods of time, preferably from one stage of differentiation to another. Given the authors' interesting observation that burst amplitude ramps up at intermediate stages of differentiation, it would be nice to see how this plays out in a single cell. Note the imaging wouldn't have to be continuous. For example, a single cell could be imaged continuously for 30 minutes, then imaged very infrequently (or not at all) for a relatively long period of time, then imaged continuously again for another 30 minutes.

This would be a very nice experiment. Unfortunately, at present, we do not think it would be feasible. First, our *in-vitro* system relies upon the development of embryoid bodies which provide the environment for continued differentiation. For live imaging, erythroid cells must be released from these cultures by trypsin-mediated disaggregation. We have not tested whether such cells are viable in the long-term or would proceed normally throughout differentiation. It would require an extensive amount of work to develop such a system. Second, although our cells are tethered to the imaging dishes by poly-L-lysine, erythroid cells are inherently suspension cells and therefore, over a long period of time, there would likely be considerable movement. Tracking might be possible, but again, a significant amount of work would be required to develop and integrate such analysis.

4. The authors develop a model based on their observations in Figure 6, but do not attempt to test the model. The study as it stands is just observational. It would be nice to perform some sort of perturbative experiments to test a model in a meaningful way. For example, siRNA could be used to knock down factors known to promoter enhancer/promoter crosstalk, such as P300 or cohesion, or activate gene expression, such as a specific transcription factor. Such perturbative experiments would yield insight into the mechanism of bursting at this gene, which would be very informative.

We agree with the referee that the model goes beyond the current data and we have consequently modified Figure 6. We do not want to give the impression that we have explained the mechanism rather than proposed a plausible hypothesis. The additional experiments proposed would certainly be important but would involve a completely new set of extensive experiments which we think would be beyond the scope of the current work.

5. The data are not particularly convincing that live cell microscope imaging meaningfully supports identification of 6 layers of differentiation strata, especially since at least 3 categories rely solely on a narrow range of CD71 fluorescence intensity. This range of CD71 expression could encompass noise in CD71 expression itself, unrelated to differentiation status.

We apologise that our explanation of this in the text was confusing. We are not suggesting that there are six discrete layers of differentiation which cells pass through during erythropoiesis. We have arbitrarily grouped cells into progressive 'stages' along the erythroid differentiation pathway (associated with changes in the levels of CD71 and Ter119 cell surface markers) to analyse how α -globin transcription changes during this process. So, the 'stages' here represent progression through a continuous process of differentiation, rather than discrete states or cell types during erythropoiesis. We have modified the text to clarify this (page 7). Also, in the

discussion we already acknowledged that our approach provides a coarse dissection of erythropoiesis (page 13).

6. The data for Fig. 5D is difficult to interpret relative to that presented in Fig. 4D. A similar presentation to that for Fig. 4D, perhaps with additional separate panels for 'low ON' vs 'high ON', would make it easier to evaluate these data.

We thank the reviewer for pointing out this potentially confusing issue and this very helpful suggestion. We agree that it initially appears confusing to use the Fano factor (as in Fig. 4D) for looking at variability in transcriptional bursting and analysing another aspect of the process by calculating the 'amplitude residuals' (as in Fig. 5). We did this because while the Fano factor is able to measure variability in transcription in general, in Figure 5 we wanted to understand specifically whether the burst amplitude was important for the changes we observed in Figure 4. To be clear, Figures 4D (and E) use the Fano factor to assess changes in intra-cellular transcriptional variability during differentiation. Figure 5 builds on these data by exploring whether differences in the transcriptional behaviour of cells might explain the changes in intra-cellular variability of alpha-globin transcription during differentiation.

To this end, we classified cells into three types of transcriptional behaviour as shown in Figure 5B – 'basal amplitude bursting' cells, and 'high amplitude bursting' cells which are either continuously active (high ON, high amplitude) or intermittently active (low ON, high amplitude). In the original manuscript, we did this by using specific information from the transcriptional time series. We measured the length of time spent transcribing (ON duration) and the amount of RNA produced during that time (transcriptional output). Using these parameters, we were able to segregate cells according to their average burst amplitude and ON duration. The average burst amplitude was calculated by deriving the 'amplitude residual' but we acknowledge that this terminology may have caused confusion and we now refer to this as the 'basal amplitude deviation score' to make it clear that we are measuring the degree to which transcription is varying from the basal level.

Use of these two measurements (ON duration and basal amplitude deviation score) gives us a more detailed picture of the behaviour of the gene compared to a metric such as the Fano factor which is simply a (mean-normalised) measure of variance, and which doesn't specifically interrogate the bursting process.

For completeness, we tested whether the Fano factor could provide a simpler alternative to the use of the basal amplitude deviation score, for segregating cells according to their average burst amplitude. In a new Extended Data Figure 10 we compare the use of the basal amplitude deviation score and the Fano factor to separate basal and high amplitude cells. For either of these to be a good metric they must be able to identify high amplitude cells regardless of the total time spent transcribing (i.e. either 'high ON' or 'low ON'). The examples of bursting cells shown in Extended Data Figure 10A provide a useful reference point here, with the blue (high ON) and red (low ON) cells all having a similarly high average burst amplitude, but differing in their ON duration. While the basal amplitude deviation score for each of these cells is comparable (Ext. Data Fig. 10C), the Fano factor for the high ON cells (blue data points, Ext. Data Fig. 10D) is almost half that of the low ON cells (red data points, Ext. Data Fig. 10D).

This shows that the Fano factor is not an appropriate measure to separate cells based on burst amplitude as it does not enable a meaningful comparison of cells with different ON durations. Accordingly, while the proportion of 'low ON, high amplitude' cells was consistent with that

presented in the original analysis, the Fano factor was less capable of identifying ‘high ON, high amplitude’ cells compared to the basal amplitude deviation score (Ext. Data Fig. 10E and F). Therefore, we do not think that the Fano factor is appropriate for segregating cells according to the different transcriptional behaviours. Nevertheless, we have included these analyses in Extended Data Figure 10 for comparison and completeness. We have also included a paragraph discussing this analysis in the Supplementary Note.

We also acknowledge that the derivation of the basal amplitude deviation score is a somewhat detailed technical issue for the main text, and therefore have moved panels Figure 5C and D from the original manuscript to the new Extended Data Figure 10 for those wishing to look at this in detail. We have also revised the results section of the main text to clarify the presentation of these data (page 12).

7. The ‘Ter19 intensity’ x-axis for Fig. 3B and other figures require additional numeric labeling.

This has been corrected in Figure 3B, Extended Data Fig. 6C and 8A, C, E and F.

Reviewer #2. *The manuscript submitted by Jeziorska, Tunnacliffe, et al. is a major step forward in our understanding the mechanisms by which genes are switched on and off during differentiation. This paper has many strengths. The question of transcriptional regulation is fundamental to almost every field of biology, and readers in any of these fields are likely to be influenced by this work. The authors are commended for keeping the narrative focused on α -globin as a model locus, rather than shifting the focus to the erythroid system. The experiments are exquisitely controlled, and the authors go to great lengths in the text and supplemental materials to make sure that the techniques and analysis tools are explained. The figures are concise and informative. It was a pleasure to review a paper that did not contain 8 panel figures.*

We thank this reviewer for their positive comments

1. My only minor suggestions are that the authors make some mention that they are studying embryonic erythropoiesis as opposed to the more familiar definitive wave.

We did mention this in the discussion but referred to this as primitive erythropoiesis rather than embryonic. We have now clarified this by referring to “primitive (embryonic) erythropoiesis” (page 13).

2. I would favor a model slide focusing on the activity of the gene itself and leave the promoter/enhancer interactions for future work.

As discussed above (Reviewer #1, point 4) we have changed Figure 6 to address this point.

Reviewer #3 *Overall, this work is very well-done and the study of transcriptional dynamics during differentiation is novel. I do not have any major comments on the quality of the experimental results and analysis.*

1. In my opinion, the authors should substantiate the mechanistic hypotheses they make by experiments probing promoter-enhancer contacts / chromatin accessibility/hub formation, and link these to the observed changes in dynamics.

The referee raises an important set of correlative parameters which will provide a broader context in which to interpret the current data. The model discussed is largely based on our extensive previous studies of exactly these changes throughout sequential stages of erythroid differentiation. These include analysis of enhancer-promoter proximity using chromosome conformation capture (Oudelaar, Beagrie et al. Nature Communications 2020, Oudelaar et al Nature Genetics 2018), chromatin accessibility (Oudelaar, Beagrie et al. Nature Communications 2020, Anguita et al EMBO J 2004, Anguita et al Blood 2007, Hay et al Nature genetics 2016) and Francis et al PLoS One 2021), transcription factor binding (Hay et al Nature Genetics 2016 and Anguita et al EMBO J 2004) and development of a transcriptional hub (Brown et al Nature Communications 2018, Oudelaar et al Nature Genetics 2018). All of these findings are summarised in Oudelaar et al Current Opinions in Genetics and development 2021. We have modified the legend to Figure 6 and the discussion (page 15) to clarify the fact that our model of alpha globin expression during erythropoiesis is based on these previous studies rather than the current work.

2. Parameters for image stack acquisition should be completed with number and thickness of slices

This information has been added to the Methods on page 21.

3. Extended data Fig.2: The data is normalised, but for a number of panels the max value is not 1 (e.g. Runx1, Nanog), this should be corrected

The data were normalised within each time course experiment, such that the highest value within the time course would be 1. If this peak in expression occurred on different days of each experimental repeat, then the averaged value (as presented in Ext. Data Fig. 2) will not always be 1. For example, in two experiments the highest levels of Nanog expression relative to Rn18s were found in the ‘mES’ sample, while in the third it was in the ‘Day 0’ sample (when cells are in induction medium before differentiation) of the time course (see below table). Averaging of these data means that both time points have values less than 1. The raw data for individual RT-qPCR experiments can be found in the required ‘Source Data’ file. We have clarified this in the figure legend.

Nanog	mES	Day 0	Day 1	Day 2	Day 3	Day 4	Day 5	Day 6	Day 7
Expt 1	0.35	1	0.29	0.27	0.67	0.03	0.007	0.001	3x10 ⁻⁷
Expt 2	1	0.69	0.20	0.31	0.17	0.01	0.003	0.004	0.004
Expt 3	1	0.28	0.18	0.18	0.33	0.02	0.005	0.006	0.03
Average	0.78	0.66	0.22	0.26	0.39	0.02	0.005	0.004	0.01

4. Extended data Fig.3B: Why showing two examples of the same ?

In this figure panel we showed two examples of the same staining to illustrate that the less intense GFP signal in the centre of the cells really does represent the boundaries of the nucleus. We have removed one of these example panels given that the reviewer prefers a single example.

REVIEWERS' COMMENTS

Reviewer #1 (Remarks to the Author):

The revised manuscript by Jeziorska et al. largely addresses the main critiques of their previous version. New smFISH data were provided to substantiate the idea that insertion of the PP7 probe did not significantly alter transcriptional dynamics at the modified locus relative to that of wild type. Arguably, the data do appear to show that the modified PP7 locus has less overall activity relative to wild type. Though minimized, this observation was acknowledged by the authors. The number of cells analyzed for this new data should be included with the figure or in the Methods.

Reviewer #3 (Remarks to the Author):

The authors have addressed all my points, thus I feel the manuscript is ready for publication. Congratulations for this very nice piece of work !

We thank the reviewers for their time and consideration of the revised manuscript. We have summarised our response to the remaining comments below.

Reviewer #1: *The revised manuscript by Jeziorska et al. largely addresses the main critiques of their previous version. New smFISH data were provided to substantiate the idea that insertion of the PP7 probe did not significantly alter transcriptional dynamics at the modified locus relative to that of wild type. Arguably, the data do appear to show that the modified PP7 locus has less overall activity relative to wild type. Though minimized, this observation was acknowledged by the authors.*

As the reviewer describes, we have acknowledged in the text of the manuscript and the figure/legend that there is potentially a small effect on the output of the modified allele. We don't believe this affects any of the conclusions described within the paper.

The number of cells analyzed for this new data should be included with the figure or in the Methods.

These data were present in the Source Data file but we have now added these to the legend of the figure.

Reviewer #3: *The authors have addressed all my points, thus I feel the manuscript is ready for publication. Congratulations for this very nice piece of work !*

We are very grateful for the positive comments and constructive review process overall.